# Towards reliable extreme weather and climate event attribution

Omar Bellprat[1,2], Virginie Guemas[1], Francisco Doblas-Reyes[1,3] & Markus G. Donat[1,4]

Climate change is shaping extreme heat and rain. To what degree human activity has increased the risk of high impact events is of high public concern and still heavily debated. Recent studies attributed single extreme events to climate change by comparing climate model experiments where the influence of an external driver can be included or artificially suppressed. Many of these results however did not properly account for model errors in simulating the probabilities of extreme event occurrences. Here we show, exploiting advanced correction techniques from the weather forecasting field, that correcting properly for model probabilities alters the attributable risk of extreme events to climate change. This study illustrates the need to correct for this type of model error in order to provide trustworthy assessments of climate change impacts.

[1] Earth Sciences Departement, Barcelona Supercomputing Center (BSC), Carrer de Jordi Girona, 29-31, 08005 Barcelona, Spain. [2] Institute for Atmospheric and Climate Science (IAC), Swiss Federal Institute of Technology (ETH) Zurich, Universtitätsstrasse 16, 8006 Zurich, Switzerland. [3] ICREA, Pg. Lluis Companys, Barcelona 08005, Spain. [4] ARC Centre of Excellence for Climate Extremes, University of New South Wales, Sydney 2052 NSW, Australia. Correspondence and requests for materials should be addressed to O.B. (email: omar.bellprat@gmail.com)

The odds of extreme weather and climate events are changing. There is today overwhelming evidence that human activity impacts extreme heat and rain[1–4]. This finding is playing a key role in establishing public awareness of climate change[5] and in taking policy decisions for climate adaptation[6]. To what degree human influence was responsible for recent high-impact events is, therefore, of high public concern. This question is increasingly being addressed by an emerging research field designated as extreme event attribution[7–10].

Event attribution aims to estimate, whether and to what degree, natural and anthropogenic drivers have favoured the occurrence of a past event from a probabilistic point of view[10]. Single-extreme weather and climate events are unique—they happen only once in the exact same manner and their probability is. therefore. strictly speaking infinitely small, which inhibits any attribution. However, single events can be described as part of classes of events, e.g., regionally constrained exceedance of a geophysical variable over a specific threshold, for which probabilities can be attached (for instance the probability to exceed the extreme 2-m air temperature observed over Europe during the summer 2003[11]). Discerning external factors in these probabilities has been the objective of diverse event attribution methods relying on model simulations that allow to compute event probabilities in the presence or absence of external forcings[7,8].

One limitation that concerns all the state-of-the-art approaches is the model ability to reliably simulate the probabilities of extreme events and the changes thereof[12–15]. By model reliability, we mean here a specific statistical condition: simulated probabilities are said to be reliable if they truly reflect the observed frequencies (within uncertainties). For example, considering all the cases where a hot summer is simulated with 20% probability, a hot summer should also occur in 20% of these cases for a model to be considered reliable[16]. This concept of reliability originates from forecasting weather and climate where trustworthy forecast probabilities are of paramount importance for decision-making[17]. Event attribution does not aim to predict a specific event, but rather quantify how an external driver has changed its likelihood[18]. Model simulations used in event attribution are, therefore, typically evaluated for the accuracy of their mean state and different modes of variability[19], as well as their ability to reproduce the physical processes involved[20], which is generally thought to be enough for a trustworthy extreme event attribution. A small number of attribution studies assessed model reliability in the past to support or discard a statement[21,22], but they remained a clear exception in the current literature and usually did not go beyond a sole assessment of reliability for evaluation purposes.

In this study, we challenge the current practice insufficiently accounting for reliability, by demonstrating that it is not only unjustified but also carries the risk of issuing overly strong attribution statements (for extreme events). To demonstrate this, we explore to which extent climate variability and long-term response to forcing are reliably represented in model simulations commonly used for event attribution[21,23,24]. We propose subsequently a way to correct for such shortcomings, exploiting advanced model correction techniques developed in weather and near-term climate forecasting. Finally, we show its impact on event attribution statements. This study advances the current literature two-fold: by demonstrating the crucial role reliability plays in an event attribution context and, in particular, by offering an approach that can cope with potentially unreliable model simulations with the goal to provide robust attribution statements.

## Results

### The role of reliability.
Causal extreme event attribution relies on climate model experiments that discern the influence of an external factor (such as increased levels of greenhouse gases). These experiments simulate the possible evolutions of the climate system using an ensemble approach, i.e., generating several climate simulations under the same conditions, but with tiny initial perturbations to obtain a range of possible climate realisations. This ensemble approach allows to quantify the probability that a rare event occurs under given conditions[24]. In the context of attribution to climate change, two types of ensembles are required: one incorporating all observed radiative forcings (ALL, i.e., including anthropogenic greenhouse gases and aerosols in addition to natural forcings) and one counterfactual using natural forcings alone (NAT, i.e., including solar forcings and volcanic aerosols). The ensembles are carried out using coupled (ocean-atmosphere) climate models or atmosphere-only models, which are forced by observed sea–surface temperature (SST) and sea–ice concentrations[7]. The atmosphere-only approach has been the dominant approach up to date leading to several large data bases designed for extreme event attribution[21,23,24]. Both types of model approaches (coupled or atmosphere-only) simulate a nonstationary climate given that the radiative and the marine boundary forcing (in the atmosphere-only experiments) change over time. A reliable ensemble response to these forcings is fundamental as illustrated in the following example.

The example uses the case of high temperatures during Northern Hemisphere summers (June-to-August (JJA)) and warm Southern Hemisphere winters, respectively, which have been the subject of many event attribution[2–4] and physical process studies[25–27]. Figure 1a shows the summer (JJA) 2-m air temperature evolution over a grid point in Sudan in an ensemble of atmosphere-only simulations from the UK Met Office quasi-operational system (HadGEM3-A[21]). The grid point in Sudan has been chosen to illustrate the influence of marine conditions on a continental area and to select a region that is highly vulnerable to climate variability judging from recent observations. The ensemble, including anthropogenic forcings (red) undergoes a positive-temperature trend. This trend is absent in the NAT simulations (green) and, hence, attributable to human-induced climate change. Both ensembles exhibit a pronounced interannual variability that is coherent among the ensemble members due to the impact of tropical SST forcings in this region[28]. Although a similar interannual signal is reflected in the observations (black lines showing two different datasets) the model ensemble range is clearly too narrow. Indeed, the observations fall outside the model ensemble range more often than if the observations could be considered an equiprobable realisation of the model ensemble. This is illustrated with a rank histogram[16] in Fig. 2 (inset in panel a). The rank histogram counts the position of the observation amongst the ranked members of the ensemble in each year. If the observation could be considered an equiprobable member of the model ensemble, the rank histogram would be perfectly flat. On the contrary, peaks stand out at the tail ends of the diagram, which illustrates that the observation falls too often outside the model range.

One way to quantify this overconfidence objectively is to compute the mean squared differences between simulated probabilities and observed frequencies to exceed a threshold (in this case 1 in 5 years hot summers). This mean difference is widely known as the reliability component of the Brier score[16], which is shown in Fig. 2 for the entire globe (shown as 1 minus the Brier score reliability component). Central Africa (see the grid point of the example indicated) reveals to be a region of general low reliability, but other areas are also affected. The lack of reliability persists even when using a set of four different atmosphere-only models from the C20C+ project[24] (Fig. 2, panel c) or a model with large global model ensemble (100 members) used in the weather@home project (panel d)[23]. This suggests that a common

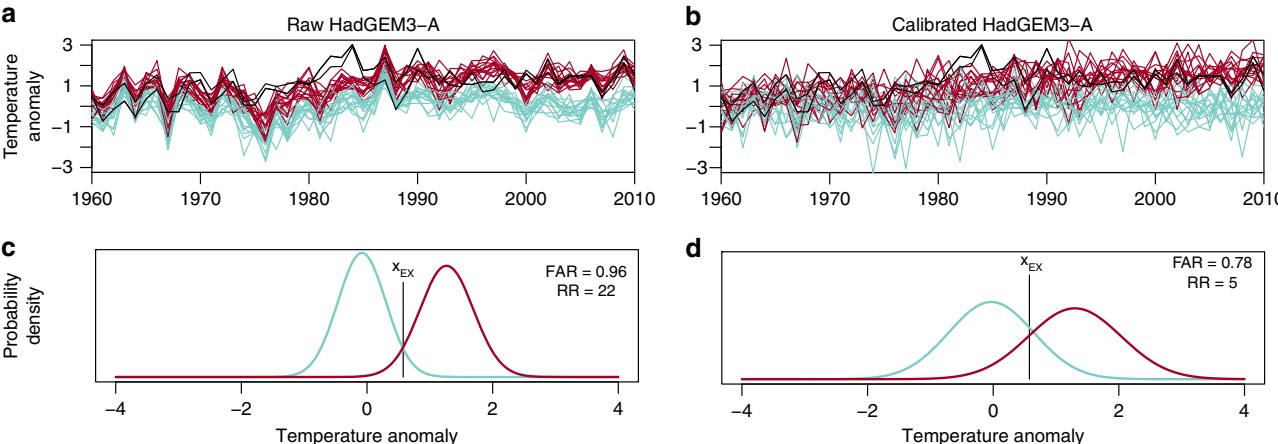

**Fig. 1** Effect of ensemble calibration on an attribution case for boreal hot summers for a grid point in Sudan (12.6°N, 34.4°W). Panels **a** and **b** show the observed historical evolution of summer temperatures (black, two different datasets) and the single-model ensemble of the UK quasi-operational attribution system (HadGEM3-A) considering all forcings (red) and only natural forcings (blue) as anomalies from present day climatology (1981–2010) derived from the all forcings ensemble. Panels **c** and **d** show the probability distribution of temperature and the associated fraction of attributable risk (FAR) due to climate change (distribution red opposed to the blue one) for a 1 in 5-year event in the NAT simulation ($x_{EX}$)

**Fig. 2** Reliability of event attribution experiments to simulate probabilities of high temperatures during boreal summers. The reliability measures the accuracy of simulated probabilities and a value of one denotes perfect reliability (see methods reliability assessment). Reliability is shown for **a** the single-model ensemble of the UK quasi-operational attribution system (HadGEM3-A), **b** the single-model system (HadGEM3-A) after the ensemble calibration, **c** the multi-model event attribution system of Climate of the 20th Century Plus project (C20C+) and **d** a 100-member ensemble of the weather@home project using HadCM3[23]. The small boxes in panels **a** and **b** denote the ranked histogram (counts of the position where the observations fall in the ensemble over the historical period) for the grid point denoted in panel **a** over Sudan analysed in Fig. 1

structural deficiency of the models or the experimental design exists that prevents the problem to be overcome by the traditional multi-model or large-ensemble approaches. The result further demonstrates that the lack of reliability (overconfidence) identified in decadal-long forced SST-driven simulations[29] does not arise from the short simulation length, but also persists in 50-year-long simulations of the C20C+ project.

Using unreliable ensembles, i.e., those in which simulated probabilities do not match the observed frequencies, to attribute a hypothetical extreme event[30] has a direct consequence on attribution statements as illustrated in Fig. 1. Assume a 1-in-5-year event (event probabilities estimated in the NAT simulation[30], same probability as used to assess the reliability) would occur in the grid-cell selected over Sudan (black vertical line) in an arbitrary recent year (2003). In the NAT ensemble this event would be much less likely than in ALL, i.e., such an event would be almost entirely attributed to human activity. To measure the level of attribution it is common to express the fraction of attributable risk[10] (fraction of attributable risk (FAR) = $1 - P_{NAT}/P_{ALL}$) or simply the risk ratio (RR = $P_{ALL}/P_{NAT}$), where $P_{NAT}$ and $P_{ALL}$ denote the probability to exceed the observed event magnitude in the NAT and ALL ensembles, respectively. Values of FAR larger than zero denote that the event is attributable to anthropogenic activity (or another external factor that is being discerned). In the example provided, FAR is almost equal to one. This means that this class of events are entirely attributable to human influence in this specific year. From visual inspection and the quantified lack of reliability, we know that the probability distributions are overly confident. The magnitude of the attributable risk is, therefore, likely overestimated and we therefore need to improve the reliability prior to the calculation of the FAR[12].

**Calibration of climate model ensembles**. Low reliability in ensemble simulations is a pervasive problem in weather and climate forecasting[31]. A range of methods referred to as ensemble calibration have therefore been proposed to overcome this deficiency[32–34]. These approaches are promising tools for the formulation of trustworthy event attribution statements, as we show in the following. Ensemble calibration corrects the model response to prevailing conditions for example as part of interannual climate variability (implicitly simulated by the model, e.g., an El Niño state) as a function of the model's ability to simulate the response to these (technically as a function of the reliability). This implies a correction of the ensemble spread (often a widening but also a narrowing is sometimes required) and a correction of the amplitude of the ensemble mean signal that deviates from the climatological state[32]. In doing so, the model ensemble becomes more reliable, i.e., more trustworthy from a probabilistic point of view.

In the context of event attribution, it is fundamental that the model response to the external driver, to which we aim to establish a causal link, is retained after the calibration. This is a concern in traditional approaches since the ensemble is simultaneously calibrated for short-term and long-term variabilities[34–36]. Here, we propose a forecast calibration technique that can cope with this difficulty. The approach relies on the ensemble inflation (see Methods) and corrects the ensemble separately for near-term variations (forced for instance by interannual SST variations or volcanic eruptions) and long-term trends (forced for instance by greenhouse gas concentration changes). Note that the climate simulations analysed here all use observed SST as ocean surface boundary conditions, which prescribes low-frequency variability in the model simulations in phase with variability in the observations. Trend differences over the 50-year period

would, therefore, primarily be due to different responses to anthropogenic forcing as opposed to climate variability. However, other more comprehensive approaches[11] would be relevant in a coupled-model setup or in presence of a highly nonlinear temporal changes in response to anthropogenic forcing. The long-term trend is only corrected for the ensemble including the anthropogenic forcings since no observations are available for a world without climate change. Also note that typical methods that correct the mean state of the simulations[37,38] (known as bias-correction methods) would not be able to correct the reliability illustrated in the example in Fig. 1 since the errors in the ensemble response differ from year to year[38].

The impact of the forecast calibration is illustrated across all three figures (Figs. 1–3). In the example provided for Sudan (Fig. 1), the calibration corrects both the ensemble mean and the spread (the long-term trend remains almost unaffected), which yields higher levels of reliability and greatly improves the rank histogram for the example (inset in Fig. 2, panel b). A lower value of FAR is obtained as a consequence. The change of FAR on a global scale is illustrated in Fig. 3. Over many regions the calibration reduces the FAR but there are also regions where FAR is underestimated by the raw model output as, e.g., Central Europe or Brazil. The overall change of FAR results from the correction of both the near-term response (ensemble correction) and the long-term change (linear trend over 50 years). The contribution from each component, as well as the impact on the risk ratio shown in the supplementary information (Supplementary Figs. 1, 2). In the example provided, correcting the near-term response particularly impacts the result over the tropics[29], while correcting the response to long-term forcing most strongly affects the mid-latitudes. The supplementary information further shows consistent results when using a different physical variable and a different return-period (JJA precipitation in a 1-in-10 year event, Supplementary Fig. 3) and an application of the calibration on artificial attribution data[12], which demonstrates that the calibration perfectly corrects the reliability and FAR in an idealised context (Supplementary Fig. 4).

While the additional analysis supports the conclusions of the study, it does not claim to be exhaustive and further applications may need to test the approach for other types of events. The current proposed method is generic, since it corrects the entire model distribution, which is advantageous compared to quantile-specific calibration approaches. However, the method might for instance not be suitable to correct a highly non-linear response to anthropogenic forcing or to correct heavily skewed native model distributions, and non-parametric approaches might be useful in such cases. Also, physical deficiencies of general circulation models as for instance the correct positioning of the storm tracks are unlikely corrected by this approach and hence calibration always be complemented with an evaluation of the model's ability to simulate the underlying physical processes[20].

## Discussion

Limitations of climate models to reliably simulate event probabilities remain overlooked in current practice of event attribution studies. Simulated probabilities can be confronted with observations as shown in this study and their reliability is a concern in simulations designed for event attribution. By exploiting techniques developed in weather and climate predictions and refining them, we have been able to improve the attribution of extreme events by taking into account the reliability of the systems used. These techniques can be refined to take into account different sources of unreliability, but it is clear that the attribution community can go a long way if it works closely with other communities concerned by this type of model error.

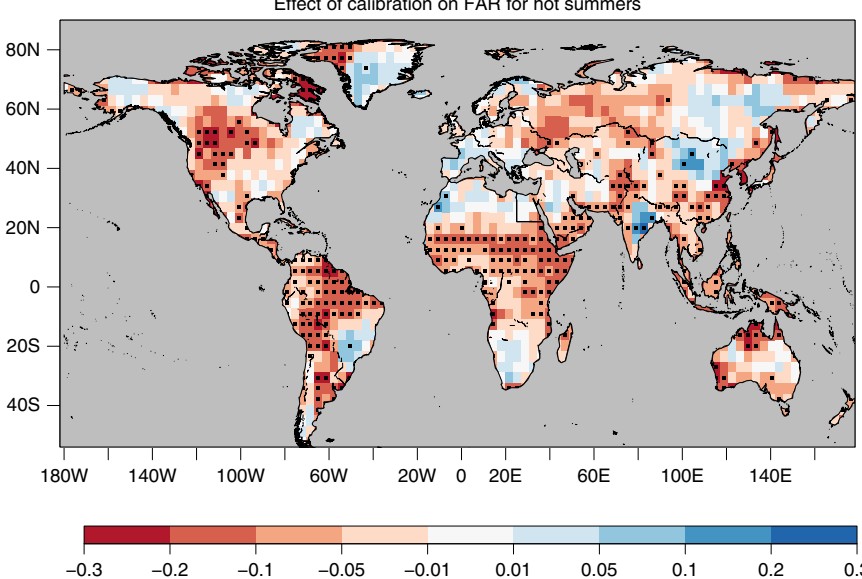

**Fig. 3** Effect of the model ensemble calibration on the fraction of attributable risk (FAR) to anthropogenic forcings of boreal hot summers. The change denotes the FAR calculated after the calibration minus the FAR calculated from the raw HadGEM3-A, as illustrated in Fig. 1 but on a global scale. The model probabilities are estimated by fitting a Normal distribution to all ensemble members. Stippling denotes a significant change in FAR determined by resampling at a 10% significance level

From a broader perspective, this study presents an example of how extreme event attribution could benefit from the experience in weather and climate forecasting. Bridging the prediction and event attribution community could enable a better understanding of the models' ability to represent the processes that lead to the occurrence of climate extremes. For instance, it has been regular practice in forecasting to disentangle drivers of a specific high impact event, using methods that allow to prescribe the observed atmospheric conditions that prevailed during the event or before it occurred[39–42]. The inability to capture past extreme events under these protocols can serve as an additional guidance on which kind of events we have sufficient confidence to carry out event attribution studies for.

It is now urgent for the scientific community to more properly address model limitations in event attribution studies because of the increasing public attention to and trust in the scientific community formulating robust attribution assessments. Adopting this practice will lead us to a more robust assessment of the human role on past extreme events for the upcoming assessment report of the Intergovernmental Panel on Climate Change (IPCC) and for the rising demand from media and international organisations such as the UN Framework Convention on Climate Change (UNFCCC) or the International Federation of the Red Cross (IFRC).

## Methods

**Model and observational data**. Atmosphere-only simulations using HadGEM3-A[21] and simulations of CAM5, MIROC5 and HadAM3P from the C20C + project[24] are used and interpolated linearly to a common resolution of 100 × 50 gridpoints. All models simulate an ensemble (15 members for HadGEM3-A and 10 members for the models in C20C+) using all radiative forcings and natural forcings only (shown in red and blue, respectively in Fig. 1). The simulations use observed SST and sea–ice conditions as boundary conditions from which a climate change signal is removed in the simulations using natural forcings alone. The observational reference used to determine the reliability and calibrate the ensemble are two-metre temperature from CRUTS3.21[43] and GHCN v3[44].

**Reliability assessment**. The ensemble reliability is measured using the reliability component of the Brier score[16]. For a given threshold (e.g., 80th quantile, 1 in 5 years event) it computes the simulated probabilities for each year, $p_k$, which is equal to the number of ensemble members that exceed the threshold divided by the total number of ensemble members. These probabilities are binned for a discrete

number of probabilities ($k = 5$ bins in this study based on quantiles of the observations and the model)[43,44]. Subsequently, the observed frequencies are computed. These consist of the occurrences of the event ($o_k$) when the $k$th probability bin was observed divided by the number of times the $k$th probability is simulated ($n_k$). The reliability is the mean squared difference between the simulated probability and the observed frequency computed for all years of the evaluation period ($N = 51$ years, 1960–2010) and summed up across all probability bins. The reliability is here expressed as 1 minus the reliability component to obtain a positive orientation of the metric (a value of 1 denoting perfect reliability[32])

$$r = 1 - \sum_{k=1}^{K} \frac{n_k}{N} \left( \frac{o_k}{n_k} - p_k \right)^2. \qquad (1)$$

The reliability is computed for the two observational datasets and then averaged to account for observational uncertainties.

**Ensemble probabilities and the FAR**. FAR compares the probability that an extreme event occurs in the ensemble using all radiative forcings ($P_{ALL}$) and the counterfactual ensemble comprising of natural forcings alone ($P_{NAT}$) as FAR = $1 - \frac{P_{NAT}}{P_{ALL}}$. The probabilities are here estimated by fitting a Normal distribution to all ensemble members. FAR is calculated in each year for the period 2001–2010 and then averaged in order to obtain a more robust signal. The difference between the calibrated and raw FAR shown in Fig. 3 is tested for statistical significance by resampling (100 iterations) with replacement all data used to estimate the probability distributions and therefore $P_{NAT}$ and $P_{ALL}$. This resampling allows to estimate a distribution of the differences between raw and calibrated FAR. When the resulting distribution of differences excludes zero at the 95th percentile the difference is stippled as significant. The influence of the reliability on FAR is also robust when using other types of distributions[12].

**Forecast calibration**. The forecast calibration relies on ensemble inflation[32], which is here expanded to correct the long-term trend and the ensemble spread individually. The recalibrated ensemble ($y$) comprises the linear trend ($x_{TR}$), the detrended ensemble mean ($x_{EM}$), and the ensemble deviation from the ensemble mean ($x'_{EM}$), while each component is multiplied by their respective correction factor that leads to the optimal calibration under Normal statistics

$$y = \alpha x_{EM} + \beta x'_{EM} + \gamma x_{TR}, \ \alpha = |\rho| \frac{\sigma_O}{\sigma_{xem}}, \ \beta = \sqrt{1 - \rho} \frac{\sigma_o}{\sigma_{x'em}}, \ \gamma = \frac{t_o}{t_x}. \qquad (2)$$

$\rho$ is the ensemble mean correlation with the observations, $\sigma_o$, $\sigma_{xem}$, and $\sigma_{x'em}$ are the standard deviation of the observations, the standard deviation of the ensemble mean, and the ensemble spread, respectively. $t_o$ and $t_x$ are respectively the linear trend slopes of the observations and the ensemble mean over the calibration period. The calibration factors are estimated from both alternative observational datasets and then averaged. We note that linear trend slopes are a relatively simple (though easily understood and therefore frequently used) measure to estimate long-term changes.

However, the assumption of linear changes may not always be justified, and more sophisticated approaches[11] to evaluate long-term changes may be useful to refine the outlined calibration approach. Also note that ideally the calibration should take into account the difference between the current and a "natural" climatology. However, in the absence of observations from a natural climate the evaluation of long-term transient changes over a period with reasonably trustworthy observations appears reasonable to calibrate the long-term climate response.

## Data availability

Model data that supports the findings of this study are available on the following repositories: HadGEM3-A, https://catalogue.ceda.ac.uk/uuid/99b29b4bfeae470599fb962?43e90cde3, C20C+: http://www.happimip.org/happi_data/, weather@home: Please consult under https://www.climateprediction.net/. Observational data sources are available on the following repositories: CRU T3.23. GHCN: ftp/ftp.ncdc.noaa.gov/pub/data/ghcn/daily/. GPCC

## Code availability

The code developed in this study is available under the following GitHub Repository: https://github.com/obellprat/NCOMMS-18-00404A. It is written using the "R" programme language and relies on the CRAN package *s2dverification*[45].

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

## Acknowledgements

This work has been supported by the EUropean CLimate and weather Events: Interpretation and Attribution (EUCLEIA) project, funded by the European Union's Seventh Framework Programme [FP7/2007–2013] under Grant agreement no. 607085 as well as the Horizon 2020 EUCP EUropean Climate Prediction system under Grant agreement no. 776613 and by the Spanish Ministry for the Economy, Industry and Competitiveness Ramón y Cajal 2017 grant reference RYC-2017-22964. We thank Antje Weisheimer and Tim Palmer from the University of Oxford for their support to conduct this research. We are further indepted to Fraser Lott, Jonas Bhend, Stefan Siegert, Karsten Haustein, and

Myles Allen for their help on the analysis and interpretation. O. Bellprat is the corresponding author (omar.bellprat@env.ethz.ch) and carried out the core analysis and writing of the paper. V. Guemas, F. Doblas-Reyes, and M. Donat have all either contributed in the study design or the interpretation of the analysis as well as the writing of the paper.

## Additional information

**Competing interests:** The authors declare no competing interests.

