## [Peer Review File · Nature Communications]

Reviewer #1 (Remarks to the Author):

Event attribution, the science of determining the change in probability of meteorological events such as heatwaves and heavy rain storms, has become a vigorous topic of research which addresses a topic with wide ranging societal implications. Over recent years, the ambition of event attribution has risen, with results being generated very rapidly by the World Weather Attribution team and a wider range of types of event being covered in the annual reports published in BAMS analysing events of the year before. But while more and more studies are being conducted, the sophistication with which model-based results are evaluated has not kept pace.

Bellprat et al propose a new method for not just evaluating event attribution assessments but for calibrating them based on observational information. This work follows on from their previous paper (Bellprat and Doblas-Reyas, 2016) which started to develop these ideas but the novelty in this work is in developing a method of calibration which potentially will produce more accurate estimates of FAR (Fraction of Attributable Risk) than non-calibrated outputs. This is a novel development worthy of publication.

I have a reservation about their technique of calibration which points to the further development of this approach rather to any fundamental flaw in what they are proposing. Their gamma term in equation 2 is a calibration on the linear trend in the model. This is a crude approximation to what is ultimately required here, which is a spatio-temporal calibration on the model's response to anthropogenic and natural forcing since pre-industrial times in order to calibrate not the linear trend but the difference between current climatology and the "natural" climatology in the absence of anthropogenic forcings. In fact, the early literature tended to do this, eg Stott, Stone, Allen Nature, 2004, first calibrated changes in European mean summer temperatures using optimal detection before calculating the changed probability of the extreme values seen in 2003. So I think as these calibration techniques are developed it is worth thinking about how best to specify gamma in equation 2. I think it would be good to refer to previous literature that does carry out this sort of partial calibration in a revised version of the paper. It would also be good to reflect in the paper on the limitations of what is proposed here. This is a step forward but not the final word.

Minor comments

Line 10 and line 16 "to which degree" -> "to what degree"

Line 20 Abstract: The upcoming IPCC report is just one reason; I think it would be better for this sentence to delete "for the upcoming IPCC report" or at least change it to make it clear that this is for a wider range of reasons.

Reviewer #2 (Remarks to the Author):

I have read the paper "Towards reliable extreme weather and climate event attribution" by O. Bellprat, V. Guemas and F. Doblas-Reyes. The core of the paper is to communicate a need for calibration of models that are used in the field of extreme event attribution. The goal is to improve the reliability of assessments on the contribution of human-induced climate change to an event. That includes improving estimates of both the magnitude of human contributions and the event's altered statistical likelihood, for instance as judged from a ratio of event frequency with and without climate change forcing. The authors discuss calibration methods for the probability methods. For various reasons enumerated below the paper is not suitable for publication, though further work may justify a re-submission.

The paper suffers from the absence of a clearly stated question that the authors seek to answer with new research. Instead, one reads at the end of the paper's first section of a "proposal" to correct for model errors germane for improving the reliability of model-based event attribution.

The paper is not novel, and the idea that model-based event attribution requires calibration is not a new proposal. For instance, Christidis et al. (2013, JCLIM) speak of the need to evaluate

modeling systems used in event attribution and they analyze reliability diagrams among other diagnostics for a set of three different extreme weather events.

In this reviewer's view, the paper presents a largely false hypothesis. It claims that a community of activity using models in event attribution judges the assessment of climatological means and modes of variability to suffice in the production of "a trustworthy extreme event attribution". The Christidis et al. example suffices to demonstrate that such a view of current practices is false and overlooks more sophisticated and advanced methods being used.

The evidence basis of the paper is weak. The authors make a sweeping statement in the paper's last section that they "have been able to improve the attribution of extreme events by taking into account the reliability of the systems used". Perhaps they are correct. However, the authors fail to provide a meaningful or comprehensive communication of the strengths and limits of their methods. Their rather facile assessment is based on a single "event", June-August seasonal temperature greater than some arbitrary threshold (a 1-in-5 year event). As further work, the paper requires additional analysis to support their claim. It would be important to know, for instance, how much altered probabilities result from calibration of the statistics of interannual variability versus an adjustment applied to the mean change (i.e., trend).

It is the reviewer's judgment that the paper itself has taken a rather careless approach to its treatment (and calibration) of long-term trends. The assumption made in the Calibration section is that an observed long-term trend (50-years) can be used to correct the ensemble mean of the anthropogenically forced model simulations. Yet, this is tantamount to assuming that a 50-year trend in observations is the anthropogenic signal, rather than being some convolution of internal noise and forced signal. Even for temperature, this assumption is problematic at a local scale.

Event attribution involves a wide spectrum of extremes, such as floods, droughts, coastal inundation, extreme extratropical cyclones, snowstorms, hurricanes, typhoons to mention a few. This reviewer is left wondering how the methods discussed in the paper would be implemented, how they would perform, and perhaps whether they would lead toward creating more reliable extreme event attribution for such situations, at least with the ease and confidence implied by the author's conclusions drawn from their example of summertime seasonal temperature.

Reviewer #3 (Remarks to the Author):

Review of "Towards reliable extreme weather and climate event attribution" by Bellprat et al.

This study proposes to apply a framework, derived from the field of weather forecasting and that allows to account for model errors, to extreme event attribution studies. The authors show that accounting for model error in this way can strongly affect the fraction of attributable risk. Given the strong reliance of event attribution on models, these results are of critical relevance for the field. The work is novel and relevant not only to scientist conducting extreme event attribution but also to a wider audience including IPCC reports' authors. The paper is well written, concise and goes straight to the point.

I strongly support its publication in Nature Communications after a few (and mostly minor) revisions, including some discussion on the advantages but also the limitations of the calibration techniques.

Major comments:

- In the description of the calibration technique (L133-143), the authors claim that ensemble calibration corrects the model response to prevailing conditions as a function of the model's ability to simulate the response to these. However, my understanding is that ensemble calibration only implicitly takes "prevailing conditions" into account via their effect on the simulated probabilities and observed frequencies of the calibrated variables for the domain of interest (here, JJA temperature in Sudan). These prevailing conditions such as the El Niño example are not explicitly taken into account (if they are and I missed it, please explain). It would be useful to clarify this point and to discuss not only what ensemble calibration can but also what it cannot do

(limitations).

Minor comments:

- The authors should improve the consistency throughout the manuscript. For instance, the authors use "pre-industrial simulations", "ensemble using only natural forcings", "NAT", etc to denote the same thing. It is fine for scientists working on event attribution, but to make the paper more accessible to a wider audience I would recommend to define one term and then used consequently throughout the manuscript (or at least in subsequent sentences).
- Figures: lettered labels are missing on all figures, and although they are sometimes referred to, they could be referred to more consistently throughout the manuscript.
- L56-58: "We use to illustrate this problem the case of hot summers in the Northern Hemisphere". Not really, the study is global and the example just lies in the Northern Hemisphere. I think the authors meant that the choice of the season is JJA, if so please reformulate (for instance: "the case of hot Northern Hemisphere summers (June-to-August, JJA), i.e. high JJA mean temperature").
- L102-105: Here the reliability is defined as the mean square difference between simulated probabilities and observed frequencies. However, the reliability shown in Fig. 2 is in fact in the opposite direction (unless I misunderstood), i.e. "one minus ..." (as also described in the Methods; L222 and Equation 1). This should be clarified in L102-105.
- L116: "a 1-in-10-year event" does not align with the caption of Figure 1, which states "a 1-in-5-year event". Also, x_{EX} is not described in the caption of Fig 1 and I only assume that it is the threshold for which FAR and RR are given (and also the 1-in-5 (or 10)-year event).
- L116-120: These two sentences are confusing: If an event (whether 1-in-5 or 1-in-10 year) occurs in the pre-industrial simulation, why would it be very unlikely on this year? Did the author mean in case the event occurs in the all-forcing/present-day/ALL simulation but does not occur in the pre-industrial simulation? Or did they mean that if an event that occurs in reality happens to correspond to a 1-in-10-year event in the NAT simulation? Please clarify (the argumentation of this paragraph could overall be improved as the main message gets somewhat lost along the way; referring more precisely to Fig 1 and the FAR/RR given there might help?).
- L121: Perhaps cite reference 10 after "risk" (i.e., before the bracket) rather than within the bracket, in order to avoid confusing it with a superscript of the equation.
- L214: It might help readers that are not familiar with the concept of reliability to add "for each year" before p_k .
- L212-227: I assume that the bins are based on the quantiles of the obs and model, respectively, but this would be useful to state.
- L168-169: An interesting feature is that the correction of the near-term response is most relevant in the tropics. It would be interesting to briefly discuss this and other features of Fig S1.
- L233-234: Please state this either in the main text or in the caption of Fig 3.
- Figure 2 (caption): refer to Methods and Equation 1 for the definition of reliability.
- Figure 3 (caption): "raw HadGEM3-A data minus FAR calculated after calibration". I believe it is the opposite, i.e. FAR after calibration minus raw FAR (negative values indicate that calibration leads to a decrease in FAR).

This document addresses the comments raised by Reviewer 1. The comments are shown in italic letters and the corresponding answers to the comments in bold letters. All the changes in the manuscript can be reviewed using tracked changes.

Event attribution, the science of determining the change in probability of meteorological events such as heatwaves and heavy rain storms, has become a vigorous topic of research which addresses a topic with wide ranging societal implications. Over recent years, the ambition of event attribution has risen, with results being generated very rapidly by the World Weather Attribution team and a wider range of types of event being covered in the annual reports published in BAMS analysing events of the year before. But while more and more studies are being conducted, the sophistication with which model-based results are evaluated has not kept pace.

Bellprat et al propose a new method for not just evaluating event attribution assessments but for calibrating them based on observational information. This work follows on from their previous paper (Bellprat and Doblas-Reyes, 2016) which started to develop these ideas but the novelty in this work is in developing a method of calibration which potentially will produce more accurate estimates of FAR (Fraction of Attributable Risk) than non-calibrated outputs. This is a novel development worthy of publication.

We thank the reviewer for her/his positive assessment of our manuscript.

I have a reservation about their technique of calibration which points to the further development of this approach rather to any fundamental flaw in what they are proposing. Their gamma term in equation 2 is a calibration on the linear trend in the model. This is a crude approximation to what is ultimately required here, which is a spatio-temporal calibration on the model's response to anthropogenic and natural forcing since pre-industrial times in order to calibrate not the linear trend but the difference between current climatology and the "natural" climatology in the absence of anthropogenic forcings. In fact, the early literature tended to do this, eg Stott, Stone, Allen Nature, 2004, first calibrated changes in European mean summer temperatures using optimal detection before calculating the changed probability of the extreme values seen in 2003. So I think as these calibration techniques are developed it is worth thinking about how best to specify gamma in equation 2. I think it would be good to refer to previous literature that does carry out this sort of partial calibration in a revised version of the paper. It would also be good to reflect in the paper on the limitations of what is proposed here. This is a step forward but not the final word.

We agree that the proposed methodology is not the final word and that the linear trend is only a first order approximation. The separation of natural and anthropogenic signals could be done in a more sophisticated way using optimal fingerprinting without the assumption of a linear trend, or other non-parametric approach that describes the forced climate signal.

In the revised version of the manuscript we discuss the limitations that arise from this assumption and refer to other literature that, in combination with the proposed forecast calibration, could lead to further more comprehensive corrections.

Lines 209 - 215:

Note that the climate simulations analysed here all use observed SST as ocean surface boundary conditions, which prescribes low frequency variability in the model simulations in phase with variability in the observations. Trend differences over the 50 year period would therefore primarily be due to different responses to anthropogenic forcing as opposed to climate variability. However, other more comprehensive

approaches¹¹ would be relevant in a coupled-model setup or in presence of a highly non-linear response to anthropogenic forcing.

We have further added the limitation of using a linear trend in the methods section.

Lines 371 - 388:

We note that linear trend slopes are a relatively simple (though easily understood and therefore frequently used) measure to estimate long-term changes. However the assumption of linear changes may not always be justified, and more sophisticated approaches¹¹ to evaluate long-term changes may be useful to refine the outlined calibration approach. Also note that ideally the calibration should take into account the difference between the current and a “natural” climatology. However, in the absence of observations from a natural climate the evaluation of long-term transient changes over a period with reasonably trustworthy observations appears reasonable to calibrate the long-term climate response.

Minor comments:

Line 10 and line 16 “to which degree” -> “to what degree”

Changed the wording - thank you.

Line 20 Abstract: *The upcoming IPCC report is just one reason; I think it would be better for this sentence to delete “for the upcoming IPCC report” or at least change it to make it clear that this is for a wider range of reasons.*

Agreed, the reference to the IPCC report has been removed.

This document addresses the comments raised by Reviewer 2. The comments are shown in italic letters and the corresponding answers to the comments in bold letters. All the changes in the manuscript can be reviewed using tracked changes.

I have read the paper "Towards reliable extreme weather and climate event attribution" by O. Bellprat, V. Guemas and F. Doblas-Reyes. The core of the paper is to communicate a need for calibration of models that are used in the field of extreme event attribution. The goal is to improve the reliability of assessments on the contribution of human-induced climate change to an event. That includes improving estimates of both the magnitude of human contributions and the event's altered statistical likelihood, for instance as judged from a ratio of event frequency with and without climate change forcing. The authors discuss calibration methods for the probability methods. For various reasons enumerated below the paper is not suitable for publication, though further work may justify a re-submission.

The paper suffers from the absence of a clearly stated question that the authors seek to answer with new research. Instead, one reads at the end of the paper's first section of a "proposal" to correct for model errors germane for improving the reliability of model-based event attribution.

The introduction was indeed lacking a clear statement of the objective of the study. The following sentence has been added describing the objective of this study.

Lines 61 - 71:

In this study, we challenge the current practice insufficiently accounting for reliability, by demonstrating that it is not only unjustified but also carries the risk of issuing overly strong attribution statements (for extreme events). To demonstrate this, we explore to which extent climate variability and long-term response to forcing are reliably represented in model simulations commonly used for event attribution^{25 - 27}. We propose subsequently a way to correct for such shortcomings, exploiting advanced model correction techniques developed in weather and near-term climate forecasting. Finally, we show its impact on event attribution statements. This study advances the current literature two-fold: by demonstrating the crucial role reliability plays in an event attribution context and, in particular, by offering an approach that can cope with potentially unreliable model simulations with the goal to provide robust attribution statements.

The paper is not novel, and the idea that model-based event attribution requires calibration is not a new proposal. For instance, Christidis et al. (2013, JCLIM) speak of the need to evaluate modeling systems used in event attribution and they analyze reliability diagrams among other diagnostics for a set of three different extreme weather events.

We respectfully disagree with the reviewer on this assessment, which also seems in stark contrast to assessments by the other reviewers who both highlight the novelty and relevance of this work.

While the reviewer is right that Christidis et al. (2013) have analysed reliability in order to evaluate their model simulations to be suitable for attribution studies, they discuss this as a measure as to whether attribution should or should not be carried out. Our study goes well beyond of what was presented by Christidis et al. (2013), because we propose an approach to calibrate the potentially unreliable model ensembles, which we show to improve reliability, and we also show the effect that the calibration for reliability has on the estimates for FAR. We would also like to highlight that Christidis et al. (2013) is a clear exception in the current literature, since very few other studies have considered the aspect of reliability.

We believe that the novelty of our study in the context of previous work has not been clear in our original submission, and have added the following sentences how our work relates to previous studies in the text

Lines 54 - 61:

A small number of attribution studies assessed model reliability in the past to support or discard a statement^{21,22}, but they remained a clear exception in the current literature and usually did not go beyond a sole assessment of reliability for evaluation purposes.

...

This study advances the current literature two-fold: by demonstrating the crucial role reliability plays in an event attribution context and, in particular, by offering an approach that can cope with potentially unreliable model simulations with the goal to provide robust attribution statements.

In this reviewer's view, the paper presents a largely false hypothesis. It claims that a community of activity using models in event attribution judges the assessment of climatological means and modes of variability to suffice in the production of "a trustworthy extreme event attribution". The Christidis et al. example suffices to demonstrate that such a view of current practices is false and overlooks more sophisticated and advanced methods being used.

We again respectfully disagree with this reviewer's assessment. While the Christidis et al study is a positive example (and a few others now discussed in the revised manuscript), it remains an exception: the majority of the attribution community does not take reliability into account - as can be easily seen when looking through the annual BAMS supplements attributing a selection of recent climate events.

Furthermore, as noted above, our study goes beyond the sole analysis of reliability. Importantly, and in contrast to previous work only recommending to check reliability as part of evaluation to decide whether or not attribution should be performed, we provide a way forward by proposing a calibration that is demonstrated to improve reliability.

The evidence basis of the paper is weak. The authors make a sweeping statement in the paper's last section that they "have been able to improve the attribution of extreme events by taking into account the reliability of the systems used". Perhaps they are correct. However, the authors fail to provide a meaningful or comprehensive communication of the strengths and limits of their methods. Their rather facile assessment is based on a single "event", June-August seasonal temperature greater than some arbitrary threshold (a 1-in-5 year event). As further work, the paper requires additional analysis to support their claim. It would be important to know, for instance, how much altered probabilities result from calibration of the statistics of interannual variability versus an adjustment applied to the mean change (i.e., trend).

We have added analysis and discussion to clarify the effects from adjusting inter-annual variability and long-term trend alone, demonstrate applicability of our suggested approach to another event type, and discuss the strengths and limitations of the approaches. The following section can be found at the end of chapter 2.

Lines 233 -266:

The contribution from each component, as well as the impact on the risk ratio shown in the supplementary information (Fig. S1, S2). In the example provided, correcting the near-term response particularly impacts the result over the tropics²⁸, while correcting the response to long-term forcing most strongly affects the mid-latitudes. The supplementary information further shows consistent results when using a different physical variable and a different return-period (JJA precipitation in a 1-in-10 year event, Fig. S3) and an application of the

calibration on artificial attribution data¹², which demonstrates that the calibration perfectly corrects the reliability and FAR in an idealized context (Fig. S4).

While the additional analysis supports the conclusions of the study, it does not claim to be exhaustive and further applications may need to test the approach for other types of events. The current proposed method is generic, since it corrects the entire model distribution, which is advantageous compared to quantile-specific calibration approaches. However, the method might for instance inappropriate to correct a highly non-linear response to anthropogenic forcing or to correct heavily skewed native model distributions, and non-parametric approaches might be useful in such cases. Also, physical deficiencies of general circulation models as for instance the correct positioning of the storm tracks are unlikely corrected by this approach and hence it should always be complemented with a evaluation of the model's ability to simulate the underlying physical processes²⁰.

It is the reviewer's judgment that the paper itself has taken a rather careless approach to its treatment (and calibration) of long-term trends. The assumption made in the Calibration section is that an observed long-term trend (50-years) can be used to correct the ensemble mean of the anthropogenically forced model simulations. Yet, this is tantamount to assuming that a 50-year trend in observations is the anthropogenic signal, rather than being some convolution of internal noise and forced signal. Even for temperature, this assumption is problematic at a local scale.

We would like to note that this analysis is based on (SST-driven) atmosphere-only simulations. This means that all models have the same oceanic boundary conditions as the observed climate, so have the same phase and representation of low-frequency variability. Assuming that internal variability on multi-decadal timescales primarily comes from the ocean, this suggests that trend differences on the ~50-year scale would be primarily due to different responses to forcing as opposed to climate variability. We have added some discussion to clarify this point in lines 209 - 215 and lines 371 - 388.

The reviewer is right, however, that different (and more sophisticated) approaches to treat long-term changes could be taken, and would in particular be recommended if fully coupled climate simulations were used. We have clarified this in the revised manuscript.

Event attribution involves a wide spectrum of extremes, such as floods, droughts, coastal inundation, extreme extratropical cyclones, snowstorms, hurricanes, typhoons to mention a few. This reviewer is left wondering how the methods discussed in the paper would be implemented, how they would perform, and perhaps whether they would lead toward creating more reliable extreme event attribution for such situations, at least with the ease and confidence implied by the author's conclusions drawn from their example of summertime seasonal temperature.

We have extended the analysis by providing an additional example of the calibration based on the case of summer precipitation which confirms the same conclusion (Fig. S4). However, as noted in the revised version of the paper we would like to highlight that this study does not claim to be exhaustive in terms of all possible applications but make the case that reliability matters for attribution studies and that techniques exist that can cope with this shortcoming. The primary goal of this paper is to outline the use of such correction techniques for event attribution studies, and we do this on the example of warm seasonal temperatures, but have also added one more example now to demonstrate there are other possible applications.

This document addresses the comments raised by Reviewer 1. The comments are shown in italic letters and the corresponding answers to the comments in bold letters. All the changes in the manuscript can be reviewed using tracked changes.

This study proposes to apply a framework, derived from the field of weather forecasting and that allows to account for model errors, to extreme event attribution studies. The authors show that accounting for model error in this way can strongly affect the fraction of attributable risk. Given the strong reliance of event attribution on models, these results are of critical relevance for the field. The work is novel and relevant not only to scientist conducting extreme event attribution but also to a wider audience including IPCC reports' authors. The paper is well written, concise and goes straight to the point.

I strongly support its publication in Nature Communications after a few (and mostly minor) revisions, including some discussion on the advantages but also the limitations of the calibration techniques.

We thank the reviewer for her/his positive assessment of our manuscript.

Major comments:

- In the description of the calibration technique (L133-143), the authors claim that ensemble calibration corrects the model response to prevailing conditions as a function of the model's ability to simulation the response to these. However, my understanding is that ensemble calibration only implicitly takes "prevailing conditions" into account via their effect on the simulated probabilities and observed frequencies of the calibrated variables for the domain of interest (here, JJA temperature in Sudan). These prevailing conditions such as the El Niño example are not explicitly taken into account (if they are and I missed it, please explain). It would be useful to clarify this point and to discuss not only what ensemble calibration can but also what it cannot do (limitations).

Indeed, the effects of ENSO or other natural forcings are taking into account implicitly as they are being modelled by the simulations. The calibration techniques corrects the ensemble response if observations indicates that the model response is too overconfident to these.

We have adapted the mentioned section to clarify this point. We have further extended the discussion on the strength and limitations of the proposed calibration approach.

Lines 257 - 266

While the additional analysis supports the conclusions of the study, it does not claim to be exhaustive and further applications may need to test the approach for other types of events. The current proposed method is generic, since it corrects the entire model distribution, which is advantageous compared to quantile-specific calibration approaches. However, the method might for instance inappropriate to correct a highly non-linear response to anthropogenic forcing or to correct heavily skewed native model distributions, and non-parametric approaches might be useful in such cases. Also, physical deficiencies of general circulation models as for instance the correct positioning of the storm tracks are unlikely corrected by this approach and hence it should always be complemented with a evaluation of the model's ability to simulate the underlying physical processes²⁰.

Minor comments:

- The authors should improve the consistency throughout the manuscript. For instance, the authors use "pre-industrial simulations", "ensemble using only natural forcings", "NAT", etc to denote the same thing. It is fine for scientists working on event attribution, but to make the paper more

accessible to a wider audience I would recommend to define one term and then used consequently throughout the manuscript (or at least in subsequent sentences).

Thank you for pointing this out, we have changed all references to NAT and ALL after introducing the acronyms.

- Figures: lettered labels are missing on all figures, and although they are sometimes referred to, they could be referred to more consistently throughout the manuscript.

Letters have been added

- L56-58: “We use to illustrate this problem the case of hot summers in the Northern Hemisphere”. Not really, the study is global and the example just lies in the Northern Hemisphere. I think the authors meant that the choice of the season is JJA, if so please reformulate (for instance: “the case of hot Northern Hemisphere summers (June-to-August, JJA), i.e. high JJA mean temperature”).

OK, we changed the wording accordingly.

- L102-105: Here the reliability is defined as the mean square difference between simulated probabilities and observed frequencies. However, the reliability shown in Fig. 2 is in fact in the opposite direction (unless I misunderstood), i.e. “one minus ...” (as also described in the Methods; L222 and Equation 1). This should be clarified in L102-105.

Indeed, this is correct. The figure shows the one minus the reliability component of the Brier score as done in Doblas-Reyes et al. (2005). This has now been clarified in the text.

- L116: “a 1-in-10-year event” does not align with the caption of Figure 1, which states “a 1-in-5-year event”. Also, x_{EX} is not described in the caption of Fig 1 and I only assume that it is the threshold for which FAR and RR are given (and also the 1-in-5 (or 10)-year event).

Thank you for this indication, a 1-in-5 year event is shown, this has been corrected in the manuscript.

- L116-120: These two sentences are confusing: If an event (whether 1-in-5 or 1-in-10 year) occurs in the pre-industrial simulation, why would it be very unlikely on this year? Did the author mean in case the event occurs in the all-forcing/present-day/ALL simulation but does not occur in the pre-industrial simulation? Or did they mean that if an event that occurs in reality happens to correspond to a 1-in-10-year event in the NAT simulation? Please clarify (the argumentation of this paragraph could overall be improved as the main message gets somewhat lost along the way; referring more precisely to Fig 1 and the FAR/RR given there might help?).

The paragraph has been adapted for clearer readability.

- L121: Perhaps cite reference 10 after “risk” (i.e., before the bracket) rather than within the bracket, in order to avoid confusing it with a superscript of the equation.

Thank you for this suggestion, it has been adopted.

- L214: It might help readers that are not familiar with the concept of reliability to add “for each year” before p_k .

Has been added.

- L212-227: I assume that the bins are based on the quantiles of the obs and model, respectively, but this would be useful to state.

Indeed, this has been added.

- L168-169: *An interesting feature is that the correction of the near-term response is most relevant in the tropics. It would be interesting to briefly discuss this and other features of Fig S1.*

We have added a section in the manuscript that discusses this aspect in more detail.

- L233-234: *Please state this either in the main text or in the caption of Fig 3.*

The sentence has been added to the caption of Fig. 3.

- *Figure 2 (caption): refer to Methods and Equation 1 for the definition of reliability.*

OK, has been added

- *Figure 3 (caption): "raw HadGEM3-A data minus FAR calculated after calibration". I believe it is the opposite, i.e. FAR after calibration minus raw FAR (negative values indicate that calibration leads to a decrease in FAR).*

Indeed, there was a mistake here, it has been corrected.